# Bee Venom and Its Peptide Component Melittin Suppress Growth and Migration of Melanoma Cells via Inhibition of PI3K/AKT/mTOR and MAPK Pathways

**DOI:** 10.3390/molecules24050929

**Published:** 2019-03-07

**Authors:** Haet Nim Lim, Seung Bae Baek, Hye Jin Jung

**Affiliations:** 1Department of Pharmaceutical Engineering & Biotechnology, Sun Moon University, 70, Sunmoon-ro 221, Tangjeong-myeon, Asan-si, Chungnam 31460, Korea; gotsla9210@naver.com; 2Eco system Lab., LOCORICO, Sun Moon University, 70, Sunmoon-ro 221, Tangjeong-myeon, Asan-si, Chungnam 31460, Korea; locorico@naver.com

**Keywords:** melanoma, bee venom, melittin, temozolomide, AKT, MAPK

## Abstract

Malignant melanoma is the deadliest form of skin cancer and highly chemoresistant. Melittin, an amphiphilic peptide containing 26 amino acid residues, is the major active ingredient from bee venom (BV). Although melittin is known to have several biological activities such as anti-inflammatory, antibacterial and anticancer effects, its antimelanoma effect and underlying molecular mechanism have not been fully elucidated. In the current study, we investigated the inhibitory effect and action mechanism of BV and melittin against various melanoma cells including B16F10, A375SM and SK-MEL-28. BV and melittin potently suppressed the growth, clonogenic survival, migration and invasion of melanoma cells. They also reduced the melanin formation in α-melanocyte-stimulating hormone (MSH)-stimulated melanoma cells. Furthermore, BV and melittin induced the apoptosis of melanoma cells by enhancing the activities of caspase-3 and -9. In addition, we demonstrated that the antimelanoma effect of BV and melittin is associated with the downregulation of PI3K/AKT/mTOR and MAPK signaling pathways. We also found that the combination of melittin with the chemotherapeutic agent temozolomide (TMZ) significantly increases the inhibition of growth as well as invasion in melanoma cells compared to melittin or TMZ alone. Taken together, these results suggest that melittin could be potentially applied for the prevention and treatment of malignant melanoma.

## 1. Introduction

Melanoma, a malignant tumor of melanocytes, is the most rapid and fatal form of skin cancer [1,2]. The onset of malignant melanoma has been reported to be influenced by various environmental and genetic factors including ultraviolet (UV) light and oncogenic BRAF mutations, respectively [3,4]. Melanoma can easily spread to other parts of the body, making treatment more difficult and causing death. Over the past 30 years, numerous therapies have been developed, but the prognosis for patients with malignant melanoma has not been improved significantly, with an average survival time of 6–9 months. The primary treatment for melanoma is surgical removal of the tumor or chemotherapy, but the treatment is incomplete and the patient’s prognosis is poor due to recurrence [5,6]. Therefore, the development of new therapies is still necessary.

The phosphatidylinositol 3-kinase (PI3K)/protein kinase B (AKT)/mammalian target of rapamycin (mTOR) and mitogen-activated protein kinase (MAPK) signaling pathways regulate cell survival, proliferation, migration and invasion, which are key functions of melanoma progression. Previous studies have shown that these pathways are constitutively activated through multiple mechanisms in malignant melanoma [7,8,9]. The sustained activation of these pathways not only protects melanoma cells from apoptosis, but also plays a decisive role in chemoresistance of melanoma. In pre-clinical models, the pharmacological inhibition of these pathways potently increases the sensitivity of melanoma cells to chemotherapeutic drugs such as cisplatin and temozolomide (TMZ). However, inhibiting only one of these pathways appears to be insufficient to effectively treat malignant melanoma [10]. Therefore, targeting PI3K/AKT/mTOR and MAPK pathways simultaneously can be a promising strategy to overcome chemoresistance of melanoma.

Bee venom (BV) is a natural toxin produced by honey bees (*Apis mellifera*) and has been widely used as a traditional medicine for a variety of diseases [11]. BV contains a variety of peptides including melittin, apamin, adolapin and enzymes, biological amines and non-peptide components [12]. Previous studies have revealed that BV possesses pain relief, anti-inflammatory, antimicrobial, antiviral and anticancer activities [13,14]. Most of all, BV’s various cancer-controlling effects such as apoptosis, necrosis, cytotoxicity and growth arrest are being found in various types of cancer cells including prostate, breast, lung, liver, skin and bladder cancers [15,16]. BV triggered proliferation inhibition and apoptotic death of melanoma cells both in vitro and in vivo experiments, implying that BV may be useful in treating melanoma.

Melittin, the major bioactive component of BV (50% of its dry weight), is a small linear peptide composed of 26 amino acids. The pharmacological efficacies of BV appear to be due primarily to the synergistic effect of melittin [17,18,19,20]. Therefore, this amphiphilic anticancer peptide may be a better choice than BV in its original form for cancer management. Although BV is known to have the anticancer activity on melanoma, the antimelanoma effect of melittin and its mode of action at the molecular level remain largely unknown. In this study, we investigated the inhibitory effect and molecular action mechanism of BV and melittin against various melanoma cells including B16F10, A375SM and SK-MEL-28. Our results showed that BV and melittin potently suppress the growth and migration of melanoma cells via dual inhibition of PI3K/AKT/mTOR and MAPK pathways, suggesting that melittin could be a promising chemotherapeutic agent for malignant melanoma treatment.

## 2. Results

### 2.1. Inhibition of Melanoma Cell Growth by BV and Melittin

To determine whether BV and melittin affect melanoma cell growth, B16F10, A375SM and SK-MEL-28 cells were treated with various concentrations (0–100 μg/mL) of BV and melittin for 72 h, and the MTT assay was performed. BV and melittin remarkably inhibited the growth of melanoma cells, with IC_50_ values of 12.67 and 4.46 μg/mL in B16F10, 5.69 and 4.43 μg/mL in A375SM, and 3.05 and 3.06 μg/mL in SK-MEL-28, respectively (Figure 1).

The growth inhibitory effect of BV and melittin against melanoma cells was further assessed using a colony formation assay. Treatment with BV and melittin resulted in a significant inhibition of the colony-forming ability in B16F10, A375SM and SK-MEL-28 melanoma cells (Figure 2). In particular, the inhibitory effect of melittin on the clonogenic growth was more potent than that of BV at the indicated concentrations. These results suggest that melittin may be considered a novel anticancer agent with potent antiproliferative activity against melanoma cells.

### 2.2. The Inhibitory Effect of BV and Melittin on Melanoma Cell Migration and Invasion

To assess the ability of BV and melittin to suppress the metastasis of melanoma cells, the effect of BV and melittin on the melanoma cell migration and invasion was evaluated by wound healing and Matrigel invasion assays, respectively. The wound scratch in untreated control cells was fully closed after 24 h of incubation, whereas treatment with BV and melittin led to the suppression of B16F10, A375SM and SK-MEL-28 melanoma cell migration in a dose-dependent manner (Figure 3).

BV and melittin also decreased the invasion of A375SM and SK-MEL-28 cells when compared with controls (Figure 4). Notably, melittin more effectively inhibited the metastatic potential of melanoma cells than BV.

### 2.3. The Antimelanogenic Effect of BV and Melittin

Malignant melanocytes tend to exhibit upregulated melanogenesis and defective melanosomes [21,22]. To investigate the effect of BV and melittin on melanogenesis of B16F10 cells, we thus determined the melanin content. The cells were stimulated by α-MSH in the presence or absence of BV and melittin for 72 h. Treatment with BV and melittin dose-dependently downregulated the melanin formation of B16F10 cells induced by α-MSH, indicating that they inhibit the melanogenesis of melanoma cells (Figure 5).

### 2.4. The Effect of BV and Melittin on Melanoma Cell Apoptosis

To further elucidate the anticancer effect of BV and melittin in melanoma cells, cellular apoptosis was quantitatively analyzed by flow cytometry following Annexin V-FITC and PI dual labeling. When melanoma cells were treated with BV and melittin for 24 h, the total amount of early and late apoptotic cells markedly increased in comparison with controls (from 0.99 to 7.80 and 46.45% for BV and melittin in B16F10 cells, from 1.18 to 35.45 and 98.60% in A375SM cells, and from 4.36 to 25.84 and 90.30% in SK-MEL-28 cells, respectively) (Figure 6). In addition, the ability of melittin to induce apoptosis was superior to BV.

Subsequently, the effect of BV and melittin on caspase activity was assessed to determine whether they induce caspase-dependent apoptosis in melanoma cells. Western blot analysis showed that treatment with BV and melittin resulted in the increased expression of cleaved caspase-3 and -9 in A375SM cells (Figure 7). Taken together, these data imply that the growth suppression of melanoma cells by BV and melittin is partly due to increased apoptosis in a caspase-dependent manner.

### 2.5. The Effect of BV and Melittin on the Regulation of PI3K/AKT/mTOR and MAPK Pathways

The oncogenic PI3K/AKT/mTOR and MAPK signaling pathways regulate cell survival, proliferation, migration and invasion, which are key features of malignant melanoma progression [7]. The effect of BV and melittin on the PI3K/AKT/mTOR and MAPK activation was therefore investigated in melanoma cells. They significantly suppressed the phosphorylation of PI3K, AKT and mTOR as well as MAPKs such as extracellular signal-regulated kinase (ERK) and p38 in A375SM cells (Figure 8A).

Particularly, the inhibitory effect of melittin on the signaling pathways was better than that of BV. However, melittin also reduced the total protein levels of these signaling molecules. To further assess whether melittin affects their proteasomal degradation, we investigated the inhibitory effect of a proteasome inhibitor MG132 on the degradation of the proteins by the activity of melittin. Treatment with MG132 abolished the degradation of PI3K, AKT, mTOR and ERK proteins by melittin, indicating that melittin induces the proteolysis of these signaling effectors in melanoma cells (Figure 8B). Therefore, the suppressive effect of melittin on the PI3K/AKT/mTOR and MAPK signaling may be partly associated with its inhibitory effect on the protein stability of such signaling molecules. Taken together, these results suggest that BV and melittin inhibit the growth and metastatic potential of melanoma cells by downregulating both PI3K/AKT/mTOR and MAPK signaling pathways. As a result, the inhibition of these signaling pathways led to a reduction in the expression of microphthalmia-associated transcription factor (MITF), which is an important regulator of melanogenesis and malignant melanoma development, as well as matrix metalloproteinase (MMP)-2 and MMP-9, which play a critical role in melanoma metastasis (Figure 8A) [23,24].

### 2.6. The Effect of a Combination of Melittin with TMZ on Melanoma Cell Growth and Invasion

Temozolomide (TMZ) is used as a chemotherapeutic agent in many carcinomas including melanoma, but its anticancer activity is insufficient due to tumor chemoresistance [25,26]. In order to further develop a promising combination therapy of melittin for malignant melanoma, we investigated the effect of melittin on chemosensitivity of melanoma cells to TMZ.

As shown in Figure 9A, combination of melittin with TMZ markedly increased growth inhibition in both A375SM and SK-MEL-28 melanoma cell lines compared with melittin or TMZ alone (inhibition rates of 42.51, 48.62 and 96.02% with 2.5 μg/mL melittin, 50 μM TMZ and melittin/TMZ combination in A375SM cells and 44.90, 39.38 and 90.32% in SK-MEL-28 cells, respectively). In addition, the combination treatment of melittin with TMZ greatly increased invasion inhibition compared with single agent treatment (inhibition rates of 51.90, 30.51 and 89.48% with 2.5 μg/mL melittin, 50 μM TMZ and melittin/TMZ combination in A375SM cells and 54.76, 36.50 and 94.95% in SK-MEL-28 cells, respectively) (Figure 9B). These findings suggest that melittin might be used to treat melanoma in combination with TMZ.

## 3. Discussion

Melanoma is one of the malignant tumors whose incidence and mortality are increasing worldwide and is notably resistant to conventional chemotherapeutic agents [1,2]. Through our continuing efforts in searching potential anticancer agents from natural products, the antimelanoma effect and underlying molecular mechanism of melittin, a main ingredient of bee venom (BV), were newly found in this study.

Melittin is an amphiphilic peptide of linear structure composed of 26 amino acids (GIGAVLKVLTTGLPALISWIKRKRQQ) in which the N-terminal part is hydrophobic, whereas the C-terminal part is hydrophilic and strongly basic [27]. Recent studies have revealed that melittin exhibits potent antitumor effect against several human cancer cell lines including ovarian, liver, cervical, bladder, gastric and breast cancers. Melittin induced apoptosis of human ovarian cancer cells via increase of death receptor expression and inhibition of signal transducer and activator of transcription 3 (STAT3) pathway [28]. On the other hand, the apoptotic effect of melittin on human gastric cancer cells was associated with activation of mitochondrial signaling pathway but not death receptor-mediated pathway [29]. Besides, melittin inhibited growth of human hepatoma cells through HDAC2-mediated PTEN upregulation and inhibition of PI3K/AKT signaling pathway [30]. Furthermore, melittin showed anticancer and antiangiogenic effects by suppressing expression of hypoxia-inducible factor-1α (HIF-1α) and vascular endothelial growth factor (VEGF) through inhibition of ERK and mTOR pathways in human cervical cancer cells [31]. However, even though BV has been reported to inhibit the proliferation of malignant melanoma cells via cell cycle arrest at G1 stage and calcium-dependent apoptotic cell death [32,33], the cellular mechanisms of the antimelanoma effect of melittin remain fully unexplained.

In the present study, we found that melittin effectively inhibits the growth of melanoma cells by inducing a caspase-dependent apoptosis. Apoptosis is caused by the organic reactions of various proteins regulated by internal and external cellular pathways [34,35]. Caspase plays an important role in apoptosis and is involved in the common pathway of various apoptotic signals. Caspases are present as inactive pro-enzymes that are activated by proteolytic cleavage. Our results indicated that melittin induces apoptosis in melanoma cells through upregulation of cleaved caspase-3 and -9.

Malignant melanoma begins with a progressive disease in the local area, but as it progresses, the tumor cells begin to penetrate into the surrounding tissues. The metastatic ability of melanoma cells determines the severity of this disease and is therefore considered an important goal in malignant melanoma management [36]. We confirmed the antimetastatic potential of melittin through in vitro migration and invasion assays using melanoma cells. In particular, melittin showed stronger anticancer efficacy than BV in both proliferative and metastatic abilities of melanoma cells.

We also observed that BV and melittin have antimelanogenic activity in α-MSH-stimulating condition. The synthesis of melanin represents a major differentiated function of melanocytes. Although the main function of melanin is to protect against UV-induced damage, melanin pigment can also attenuate effectiveness of radiation or chemotherapy. In addition, melanogenesis can enhance melanoma progression owing to immunosuppressive, genotoxic and mutagenic properties [21,22,37]. In this study, treatment with melittin efficiently blocked the melanin formation of melanoma cells stimulated by α-MSH, indicating that melittin downregulates the differentiation of melanoma cells associated with melanogenesis.

Previous studies have shown that melittin downregulates PI3K/AKT/mTOR and MAPK signaling pathways, thereby leading to apoptosis in several tumor cells such as liver and cervical cancers and inhibits the metastatic ability of cancer cells [30,31]. These pathways have been also demonstrated to be important in development of malignant melanoma by promoting cell proliferation and metastasis [7]. To elucidate the molecular mechanism responsible for the malignant melanoma inhibitory effect of melittin, we confirmed whether melittin affects PI3K/AKT/mTOR and MAPK signaling. Treatment with melittin caused a reduction in the phosphorylation of PI3K, AKT and mTOR as well as MAPKs including ERK and p38 in melanoma cells. Notably, the inhibitory effect of melittin on the PI3K/AKT/mTOR and MAPK signaling was better than that of BV. Meanwhile, melittin also decreased the total protein levels of these signaling molecules, implying that the suppressive effect of melittin on the PI3K/AKT/mTOR and MAPK signaling is partly due to its inhibitory effect on the protein expression of such signaling molecules. Taken together, melittin could be a more efficient anticancer agent against malignant melanoma by simultaneously targeting the PI3K/AKT/mTOR and MAPK pathways.

Current standard therapy for malignant melanoma includes a combination of resectional surgery with chemotherapy such as temozolomide (TMZ), an alkylating agent that induces apoptosis through DNA strand breaks [25,26]. However, metastatic melanoma is highly chemoresistant and the chemotherapeutic drug yielded very limited survival benefit. Recent studies have shown that inhibition of PI3K/AKT/mTOR signaling sensitizes melanoma cells to TMZ [38,39]. We thus evaluated whether melittin ameliorates the anticancer effect of TMZ. Combined treatment with melittin and TMZ more effectively inhibited growth and invasiveness of melanoma cells compared with melittin or TMZ alone. These data suggest that melittin may have a promising therapeutic potential to overcome the chemoresistance to TMZ of melanoma patients.

Nevertheless, melittin is known to have a strong hemolytic activity [19,27]. Even though we demonstrated significant anticancer activity of melittin against malignant melanoma, further studies to reduce the nonspecific cell lysis and toxicity of melittin will be needed.

## 4. Materials and Methods

### 4.1. Materials

Melittin, temozolomide, MG132 and alpha-melanocyte stimulating hormone (α-MSH) were obtained from Sigma-Aldrich (St. Louis, MO, USA). Bee venom, Matrigel and Transwell chamber system were obtained from Chung Jin Biotech (Ansan, Korea), BD Biosciences (San Jose, CA, USA) and Corning Costar (Acton, MA, USA), respectively. Dulbecco’s modified Eagle’s medium (DMEM) and fetal bovine serum (FBS) were purchased from Invitrogen (Grand Island, NY, USA). Anti-phospho-PI3K, anti-PI3K, anti-phospho-AKT, anti-AKT, anti-phospho-mTOR, anti-mTOR, anti-phospho-ERK1/2, anti-ERK1/2, anti-phospho-p38, anti-p38, anti-cleaved caspase-3, anti-cleaved capase-9, anti-MITF, anti-MMP-2, anti-MMP-9 and anti-β-actin antibodies were purchased from Cell Signaling Technology (Beverly, MA, USA).

### 4.2. Cell Culture and Cell Growth Assay

B16F10, A375SM and SK-MEL-28 melanoma cell lines were obtained from the Korean Cell Line Bank (KCLB). All cells were grown in DMEM supplemented with 10% FBS and maintained at 37 °C in a humidified 5% CO_2_ incubator. Cell growth was examined using the 3-(4,5-dimethylthiazol-2-yl)-2,5-diphenyltetrazolium bromide (MTT) colorimetric assay. The cells were seeded in 96-well culture plates at a density of 2 × 10^3^ cells/well. After 24 h incubation, various concentrations of BV and melittin were added to each well. After 72 h, 50 µL of MTT solution (2 mg/mL; Sigma-Aldrich) was added to each well, and the cells were incubated for 3 h. To dissolve formazan crystals, the culture medium was removed and an equal volume of DMSO was added to each well. The absorbance of each well was determined at a wavelength of 540 nm using a microplate reader (Thermo Fisher Scientific, Vantaa, Finland).

### 4.3. Colony Formation Assay

Melanoma cells were seeded in 6-well culture plates at a density of 5 × 10^2^ cells/well. After 24 h incubation, the cells were treated with BV and melittin for 8–10 days. Following this, the cell colonies were fixed with 4% formaldehyde and stained with 0.5% crystal violet solution.

### 4.4. Chemoinvasion Assay

Cell invasion was assayed using Transwell chamber inserts. The lower side of the polycarbonate filter was coated with 10 µL of gelatin (1 mg/mL), and the upper side was coated with 10 µL of Matrigel (3 mg/mL). Melanoma cells (1 × 10^5^) were seeded in the upper chamber of the filter, and BV and melittin were added to the lower chamber filled with medium. The chamber was incubated for 24 h, and the cells were fixed with methanol and stained with H & E. The total number of cells that invaded the lower chamber of the filter was counted using an optical microscope (Olympus, Center Valley, PA, USA).

### 4.5. Cell Migration Assay

Melanoma cells were seeded in 24-well culture plates at a density of 2 × 10^5^ cells/well and grown to 90% confluence. The confluent monolayer cells were scratched using a pipette tip and each well was washed with phosphate-buffered saline (PBS) to remove non-adherent cells. The cells were treated with BV and melittin and then incubated for up to 24 h. The perimeter of the central cell-free zone was confirmed under an optical microscope (Olympus).

### 4.6. Apoptosis Analysis

The apoptotic cell distribution was determined using the MUSE Annexin V & Dead Cell Kit (Merck KGaA, Darmstadt, Germany) according to the manufacturer’s instructions. Briefly, after treatment with BV and melittin, all cells were collected and diluted with PBS containing 1% bovine serum albumin (BSA) as a dilution buffer to a concentration of 5 × 10^5^ cells/mL. 100 μL of Annexin V/Dead Cell reagent and 100 μL of a single cell suspension were mixed in a microtube and in the dark for 20 min at room temperature. The cells were then analyzed using the Muse Cell Analyzer (Millipore Corporation, Hayward, CA, USA).

### 4.7. Western Blot Analysis

Cell lysates were separated by 10% sodium dodecyl sulfate-polyacrylamide gel electrophoresis (SDS-PAGE), and the separated proteins were transferred to polyvinylidene difluoride (PVDF) membranes (Millipore, Billerica, MA, USA) using standard electroblotting procedures. The blots were blocked and immunolabeled with primary antibodies against phospho-PI3K, PI3K, phospho-AKT, AKT, phospho-mTOR, mTOR, phospho-ERK1/2, ERK1/2, phospho-p38, p38, cleaved capase-3, cleaved capase-9, MITF, MMP-2, MMP-9 and β-actin overnight at 4 °C. Immunolabeling was detected with an enhanced chemiluminescence (ECL) kit (Bio-Rad Laboratories, Hercules, CA, USA), according to the manufacturer’s instructions.

### 4.8. Measurement of Melanin Content

B16F10 cells (15 × 10^4^ cells/well) were plated in 12-well culture plates and then treated with BV and melittin in the presence or absence of α-MSH (200 nM) for 72 h. The cells were then washed with PBS and lysed in 150 µL of 1 M NaOH at 95 °C. The lysate was added in 96-well microplate, and the absorbance was measured at 405 nm using a microplate spectrophotometer (Thermo Fisher Scientific).

### 4.9. Statistical Analysis

The results are expressed as the mean ± standard error (SE). Student’s *t*-test was used to determine statistical significance between the control and the test groups. A *p*-value of <0.05 was considered to indicate a statistically significant difference.

## 5. Conclusions

In this study, we demonstrated that BV and its main component melittin potently suppress multiple oncogenic processes including growth, clonogenicity, migration, invasion and melanogenesis in malignant melanoma cells. Particularly, melittin showed stronger anticancer effects than BV. We also found that BV and melittin induce apoptosis in a caspase-dependent manner. Moreover, their antimelanoma effects are involved in the downregulation of PI3K/AKT/mTOR and MAPK signaling pathways. Noticeably, the combination of melittin with the chemotherapeutic agent TMZ increases sensitivity of melanoma cells towards TMZ. Based on these findings, we suggest that melittin could be an attractive candidate to treat malignant melanoma.

## Figures and Tables

**Figure 1 molecules-24-00929-f001:**
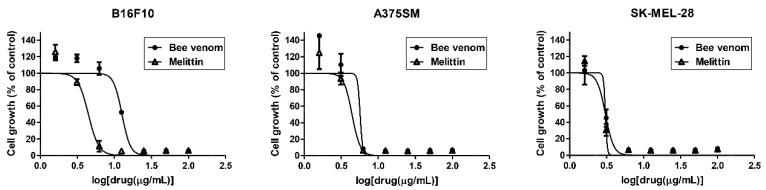
The effect of BV and melittin on the growth of melanoma cell lines. Cells were treated with various concentrations of BV and melittin for 72 h, and cell growth was measured using the MTT colorimetric assay. Data were presented as percentage relative to DMSO-treated control (% of control). Each value represents the mean ± SE from three independent experiments.

**Figure 2 molecules-24-00929-f002:**
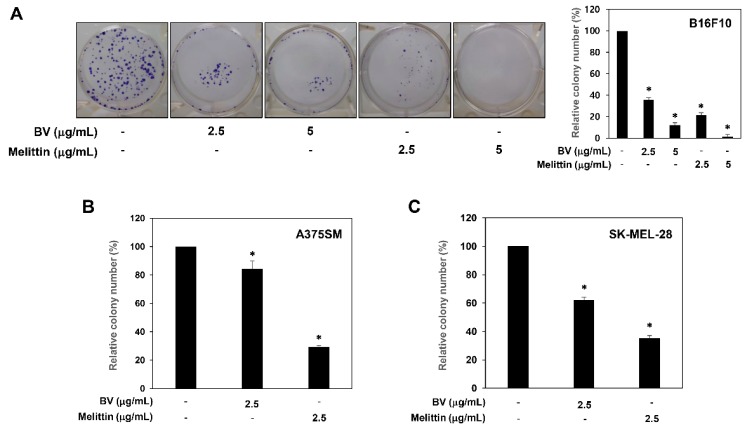
The effect of BV and melittin on the colony forming ability of melanoma cell lines. (**A**) B16F10, (**B**) A375SM and (**C**) SK-MEL-28 cells were treated with BV and melittin for 8–10 days. The cell colonies were detected by crystal violet staining and then counted. * *p* < 0.05 versus the control. Each value represents the mean ± SE from three independent experiments.

**Figure 3 molecules-24-00929-f003:**
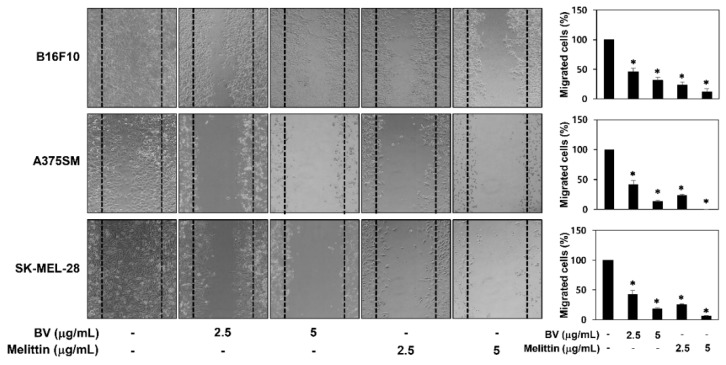
The effect of BV and melittin on the migration of melanoma cell lines. The migratory potential of melanoma cells was analyzed using wound healing assay. Cells were treated with BV and melittin for 24 h. Dotted black lines indicate the edge of the gap at 0 h. * *p* < 0.05 versus the control. Each value represents the mean ± SE from three independent experiments.

**Figure 4 molecules-24-00929-f004:**
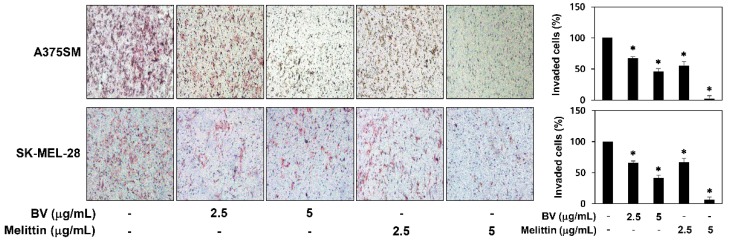
The effect of BV and melittin on the invasion of melanoma cell lines. The invasiveness of melanoma cells was analyzed using Matrigel-coated polycarbonate filters. Cells were treated with BV and melittin for 24 h. Cells penetrating the filters were stained and counted under an optical microscope. * *p* < 0.05 versus the control. Each value represents the mean ± SE from three independent experiments.

**Figure 5 molecules-24-00929-f005:**
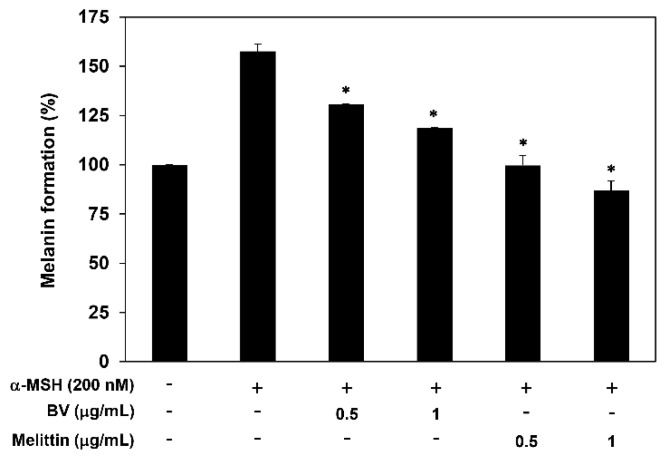
The effect of BV and melittin on the melanogenesis of α-MSH-stimulated B16F10 cells. Cells were treated with BV and melittin in the presence or absence of α-MSH for 72 h, and the cellular melanin contents were determined. * *p* < 0.05 versus the α-MSH control. Each value represents the mean ± SE from three independent experiments.

**Figure 6 molecules-24-00929-f006:**
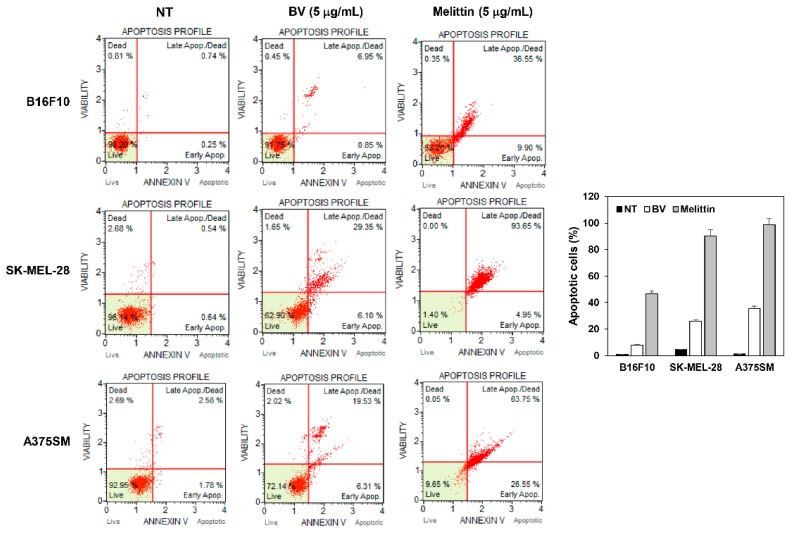
The effect of BV and melittin on the apoptotic cell death of melanoma cell lines. Cells were treated with BV and melittin for 24 h. Apoptotic cells were determined by flow cytometric analysis following annexin V-FITC and propidium iodide (PI) dual labeling. Each value represents the mean ± SE from three independent experiments.

**Figure 7 molecules-24-00929-f007:**
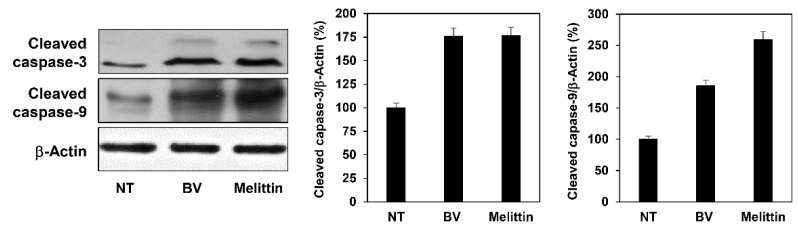
The effect of BV and melittin on the expression of apoptosis regulatory proteins in A375SM melanoma cells. Cells were treated with BV and melittin for 24 h, and the expression levels of cleaved caspase-3 and cleaved caspase-9 were detected by Western blotting. The levels of β-actin were used as an internal control. Each value represents the mean ± SE from three independent experiments.

**Figure 8 molecules-24-00929-f008:**
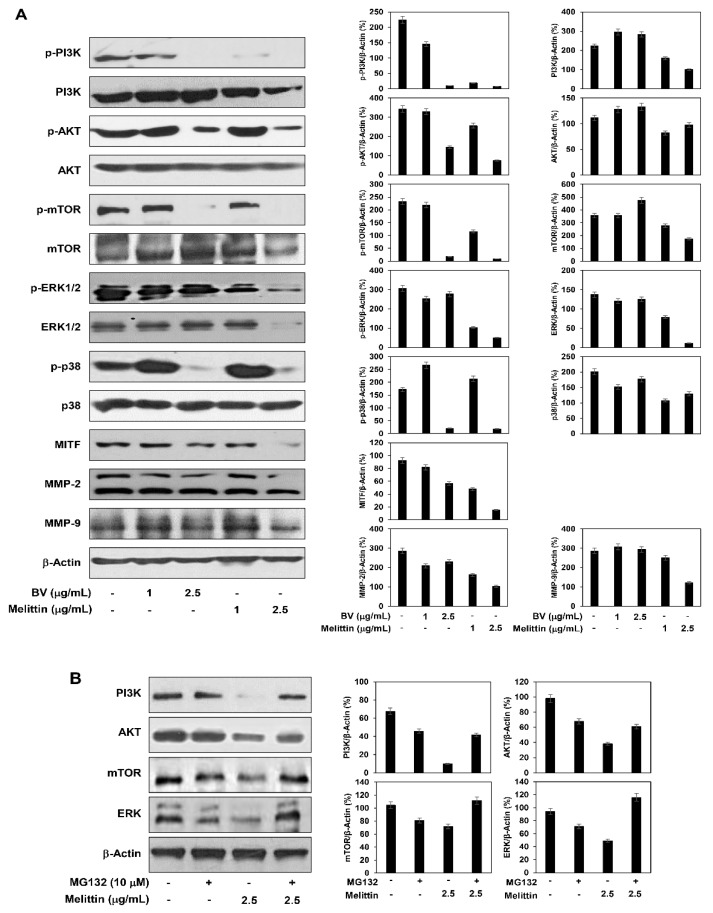
The effect of BV and melittin on the regulation of PI3K/AKT/mTOR and MAPK pathways. A375SM melanoma cells were treated with (**A**) BV, melittin and (**B**) MG132, and the protein levels were detected by Western blot analysis using specific antibodies. The levels of β-actin were used as an internal control. Each value represents the mean ± SE from three independent experiments.

**Figure 9 molecules-24-00929-f009:**
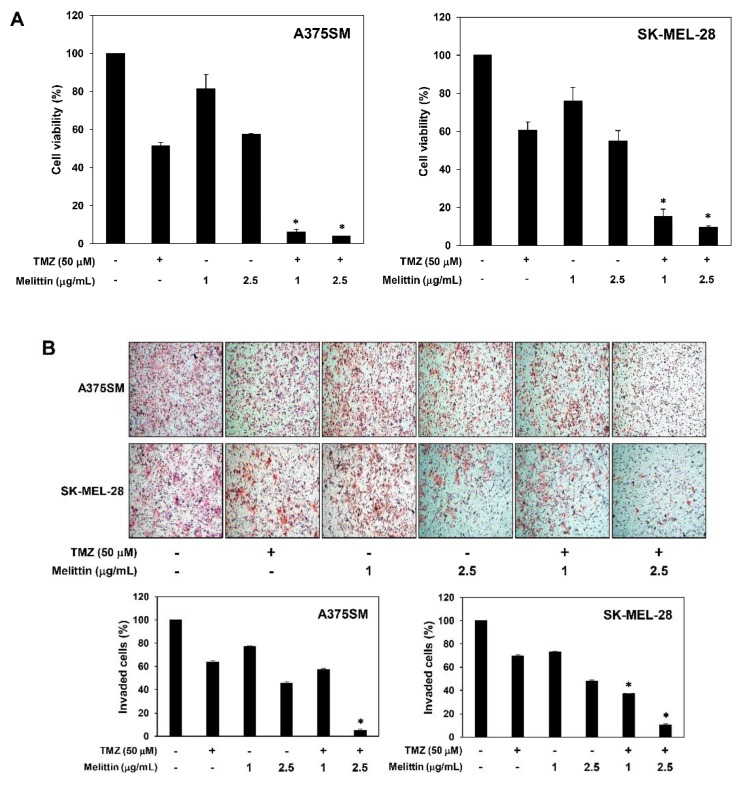
The effect of the combination of melittin with TMZ on the growth and invasion of melanoma cell lines. (**A**) Cells were treated with melittin and TMZ for 72 h, and cell growth was measured using the MTT colorimetric assay. (**B**) Cells were treated with melittin and TMZ for 24 h. Cells penetrating the Matrigel-coated polycarbonate filters were stained and counted under an optical microscope. * *p* < 0.05 versus the single agent treatment. Each value represents the mean ± SE from three independent experiments.

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
