# Peer review of "Bee Venom and Its Peptide Component Melittin Suppress Growth and Migration of Melanoma Cells via Inhibition of PI3K/AKT/mTOR and MAPK Pathways"

_molecules, 2019, doi:10.3390/molecules24050929_

Round 1

Reviewer 1 Report

In the manuscript entitled “Bee Venom and Its Peptide Component Melittin 2 Suppress Growth and Migration of Melanoma Cells 3 via Inhibition of PI3K/AKT/mTOR and MAPK 4 Pathways”, Lim and colleagues studied the effects of Melittin and BV on melanoma treatment. They found that Melittin and BV reduced the melanin formation and apoptosis of different melanoma cell lines. Furthermore, they reported that BV and melittin suppressed PI3K/AKT/mTOR and MAPK signaling pathways. Overall this study is convincing and novel, and there are some minor suggestions:

1.      for Figure 1, the two indicators are similar, so it is tough to distinguish them on the figure. The authors can consider change one indicator to a different style (such as or ).

2.      Can the authors confirm the cell lines they used are plasma negative?

Author Response

1. for Figure 1, the two indicators are similar, so it is tough to distinguish them on the figure. The authors can consider change one indicator to a different style (such as or ).

Response: According to valuable comment of the reviewer, we changed one indicator to a different style in order to clearly distinguish (¢ Þ r; Figure 1 in the revised manuscript).

2. Can the authors confirm the cell lines they used are plasma negative?

Response: To prevent the mycoplasma contamination of cultured cells, we are using the commercially available anti-mycoplasma agents for cell culture as well as disinfectant solution for incubators and sterile benches.  

Reviewer 2 Report

This paper suggests that melittin2 could be responsible for the anticancer effects of BV in melanoma and may be used as an agent for the treatment of malignant melanoma. The results are interesting and most of methods/results are well performed and well interpreted. There are some concerns and suggestions for modifications as detailed below

1. ERK negatively regulates melanogenesis in melanocytes and melanoma cells. Although BV and melittin significantly suppressed ERK phosphorylation in Fig. 8, they effectively inhibited melanin synthesis in Fig. 5. Is there different mechanism for anticancer effects of BV and melittin between melanotic and amelnotic melanoma or between mouse and human melanoma?

2. In Fig 7, the authors should perform re-immunoblot analysis for cleaved caspase-3, and -9.

3. In Fig. 8, the levels of both total and phospho forms of PI3K, AKT, mTOR, and ERK1/2 were decreased in melittin (2.5) treated cells but not by BV. Do they affect activity and stability of these signal proteins via different mechanism?

4.  Authors should show at least one mRNA or protein expression for metastasis or invasion marker genes such as MMP, TIMP etc

Author Response

1. ERK negatively regulates melanogenesis in melanocytes and melanoma cells. Although BV and melittin significantly suppressed ERK phosphorylation in Fig. 8, they effectively inhibited melanin synthesis in Fig. 5. Is there different mechanism for anticancer effects of BV and melittin between melanotic and amelnotic melanoma or between mouse and human melanoma?

Response: Although accumulating evidence has shown that ERK negatively regulates melanogenesis, PI3K and MAPK signaling can lead to increased expression of microphthalmia-associated transcription factor (MITF), a key regulator of melanogenesis and malignant melanoma development. We thus investigated the effect of BV and melittin on MITF expression. As a result, the inhibition of such signaling pathways by BV and melittin decreased the expression of MITF, suggesting that their antimelanogenic activities may be associated with the downregulation of MITF (lines 169-171; Figure 8A in the revised manuscript).

2. In Fig 7, the authors should perform re-immunoblot analysis for cleaved caspase-3, and -9.

Response: We thus performed re-immunoblot analysis for cleaved caspase-3 and -9 and changed to the obtained high-resolution images (Figure 7 in the revised manuscript).

3. In Fig. 8, the levels of both total and phospho forms of PI3K, AKT, mTOR, and ERK1/2 were decreased in melittin (2.5) treated cells but not by BV. Do they affect activity and stability of these signal proteins via different mechanism?

Response: To further assess whether melittin affects their proteasomal degradation, we investigated the inhibitory effect of a proteasome inhibitor MG132 on the degradation of the proteins by the activity of melittin. Treatment with MG132 abolished the degradation of PI3K, AKT, mTOR and ERK proteins by melittin, indicating that melittin induces the proteolysis of these signaling effectors in melanoma cells (lines 161-167; Figure 8B in the revised manuscript). Therefore, the suppressive effect of melittin on the PI3K/AKT/mTOR and MAPK signaling may be partly associated with its inhibitory effect on the protein stability of such signaling molecules.

4. Authors should show at least one mRNA or protein expression for metastasis or invasion marker genes such as MMP, TIMP etc.

Response: According to valuable comment of the reviewer, we confirmed the protein expression levels of MMP-2 and MMP-9. BV and melittin led to a reduction in the expression of MMP-2 and MMP-9, which play a critical role in melanoma metastasis (lines 171-173; Figure 8A in the revised manuscript).

Round 2

Reviewer 2 Report

No comments